# Application of big-data for epidemiological studies of refractive error

**Michael Moore** [1]*, **James Loughman**[1], **John S. Butler**[1,2], **Arne Ohlendorf**[3,4],
**Siegfried Wahl**[3,4], **Daniel I. Flitcroft**[1,5]

**1** Centre for Eye Research Ireland, School of Physics and Clinical and Optometric Sciences, Technological
University Dublin, Dublin, Ireland, **2** School of Mathematical Sciences, Technological University Dublin,
Dublin, Ireland, **3** Technology & Innovation, Carl Zeiss Vision International GmbH, Turnstrasse, Aalen,
Germany, **4** Institute for Ophthalmic Research, Center for Ophthalmology, Eberhard Karls University of
Tübingen, Elfriede-Aulhorn-Straße, Tübingen, Germany, **5** Children's University Hospital, Dublin, Ireland

* michael.moore@tudublin.ie

## Abstract

CANADA

**Data Availability Statement:** The data from this
study is available on request. This data contains
potentially identifying and sensitive patient data
such as date of birth, date of exam and county of
residence. The TU Dublin Research and Ethics

### Purpose

To examine whether data sourced from electronic medical records (EMR) and a large indus-
trial spectacle lens manufacturing database can estimate refractive error distribution within
large populations as an alternative to typical population surveys of refractive error.

### Subjects

A total of 555,528 patient visits from 28 Irish primary care optometry practices between the
years 1980 and 2019 and 141,547,436 spectacle lens sales records from an international
European lens manufacturer between the years 1998 and 2016.

### Methods

Anonymized EMR data included demographic, refractive and visual acuity values. Anon-
ymized spectacle lens data included refractive data. Spectacle lens data was separated into
lenses containing an addition (ADD) and those without an addition (SV). The proportions of
refractive errors from the EMR data and ADD lenses were compared to published results
from the European Eye Epidemiology (E3) Consortium and the Gutenberg Health Study
(GHS).

### Results

Age and gender matched proportions of refractive error were comparable in the E3 data and
the EMR data, with no significant difference in the overall refractive error distribution ($\chi^2$ =
527, p = 0.29, DoF = 510). EMR data provided a closer match to the E3 refractive error distri-
bution by age than the ADD lens data. The ADD lens data, however, provided a closer
approximation to the E3 data for total myopia prevalence than the GHS data, up to age 64.

Committee has placed restrictions on disseminating this data. Data access requests can be sent to researchethics@tudublin.ie quoting ethics approval REC-18-124.

**Funding:** Arne Ohlendorf and Siegfried Wahl are employees of Carl Zeiss Vision International GmbH. These authors provided access to some of the data used in this study and reviewed the work before submission. The funder provided support in the form of salaries for authors [AO, SW], but did not have any additional role in the study design, data collection and analysis, decision to publish, or preparation of the manuscript. The specific roles of these authors are articulated in the 'author contributions' section.

**Competing interests:** Arne Ohlendorf and Siegfried Wahl are employees of Carl Zeiss Vision International GmbH. This does not alter our adherence to PLOS ONE policies on sharing data and materials. There are no other competing interests to declare.

## Conclusions

The prevalence of refractive error within a population can be estimated using EMR data in the absence of population surveys. Industry derived sales data can also provide insights on the epidemiology of refractive errors in a population over certain age ranges. EMR and industrial data may therefore provide a fast and cost-effective surrogate measure of refractive error distribution that can be used for future health service planning purposes.

## Introduction

Refractive errors occur when the eye does not correctly focus light at the retina which results in blurred vision. It arises as a result of the eye growing too long (myopia/short sightedness), the eye not growing long enough (hyperopia/long sightedness), uneven focussing due to corneal shape (astigmatism) or a failure to focus at close ranges due to aging (presbyopia). In order to obtain clear vision, correction either through the use of optical aids such as spectacles or contact lenses or refractive surgery is required.

Refractive errors are a leading cause of vision impairment and blindness globally, due to limited access to optical correction in some regions [1], and the range of ocular diseases for which refractive errors, in particular myopia, are an identified risk factor [2,3]. There is a growing concern about myopia due to the rapid rise in global prevalence over the last few decades [4]. Vitale et al [5] found an increase in myopia prevalence from 25% in 1971–1972 to 41.6% in 1999–2004 in the United States of America. Similar increases have been observed in Europe, with higher levels of myopia observed in more recent birth cohorts [6]. The largest increases in myopia prevalence have been observed in Asia [7], particularly east Asia, with rates reaching 84% in older children [8]. The level of myopia prevalence is not as high in South America [9,10] or Africa [11], however, it is expected to rise significantly in all parts of the world in the coming years [4]. Holden et al [4] estimated that almost half of the world's population will be myopic by 2050, with almost 10% set to be highly myopic. The authors extrapolated these myopia rates by using data from published population surveys of refractive error. The primary limitation identified in this study was the significant lack of global epidemiological refractive error data, with many countries having no data whatsoever or significant gaps in data across different regions, age groups and ethnicities. The authors made specific reference to the reduced certainty with regards to their high myopia predictions, with only 48 studies contributing data to these projections.

In order to assess the public health implications of refractive errors, it is essential to have accurate population-based epidemiological data. In light of the observed differences between countries and changing prevalence over time, such data needs to be both representative of a given population and current. In Europe, epidemiological data has been collected over many decades, often from historical cohorts. The largest such study [12], the European Eye Epidemiology (E3) consortium of 33 groups from 12 European countries, collated data on 124,000 European participants from population cohort and cross-sectional studies on refractive error conducted between 1990 and 2013. While this data does show a trend of increased myopia prevalence for people born in more recent decades, the available data from recent years and on younger population cohorts is relatively sparse.

Gathering comprehensive epidemiological data that can determine global prevalence trends in refractive error over time using this traditional methodology is slow and open to question in terms of cost effectiveness [13,14]. For this reason, the growing volume of data gathered in

healthcare in recent years is of specific interest. Data such as electronic medical records (EMR) and industrial manufacturing or sales records represent a potentially valuable source of secondary data, i.e. data used for a purpose that is different from that for which it was originally collected. The scale of such data is often far larger than conventional research datasets and it is now commonly referred to as Big Data. Big Data is now recognized as an important resource for scientific research, allowing conclusions to be drawn that would otherwise be impossible using traditional scientific techniques [15,16].

In the field of eyecare, several studies have demonstrated the usefulness of EMR data for determining disease epidemiology [17,18] and treatment outcomes [19,20]. The application of such approaches to myopia genetics research has shown strong correlation with the results obtained using conventional epidemiological research methodologies [21,22]. National [23,24] and private insurance claims records have also been used to determine the epidemiology of several ocular diseases, as have hospital records [25]. Big Data sources of this type can be used as an alternative form of epidemiological data, particularly in the absence of conventional epidemiological studies. Datasets such as national insurance claims records can be generalised to an entire population while EMR and hospital record data are useful when considering specific population cohorts.

The potential of Big Data as a tool to monitor population trends in refractive error has received little attention. Optometric EMR data provides an obvious example of a rich source of data on refractive error that has yet to be exploited for this purpose. Another novel, but less obvious, source of data is the manufacturing and sales records of companies involved in the supply of optical appliances such as spectacle and contact lenses. This data source is much more limited in terms of the information available, but the ubiquity of these optical appliances indicates such data may still elicit useful insights on refractive error epidemiology.

This study was designed, therefore to examine whether optometric EMR data or spectacle lens data can provide estimates of refractive error distribution that are comparable to traditional population surveys.

## Methods

Anonymized EMR data was gathered from 28 Irish optometry practices. The data was extracted remotely through the EMR provider following provision of explicit consent from the data (practice) owners during the period of May 2018 to June 2019 for all 28 practices. This study was approved by the TU Dublin Research Ethics and Integrity Committee and adheres to the tenets of the Declaration of Helsinki (REC-18-124). Patient level consent was not required due to the nature of the anonymization of the data. The data extracted comprised all practice records since first use up to the date of extraction for each practice. The EMR provider removed any personally identifying data and anonymized the data prior to delivery so that the anonymization could not be reversed by the researchers. The data was analysed using the R programming language (R Core Team (2020). R: A language and environment for statistical computing. R Foundation for Statistical Computing, Vienna, Austria. URL https://www.R-project.org/.). At the time of extraction, a new unique identifying number was generated for each subject within the EMR data allowing their data to be tracked across multiple visits. The data available for each subject included demographic, refractive, visual acuity, binocular vision, contact lens, ocular health and clinical management data. For this analysis only demographic, refractive and visual acuity data were considered with most refractions having been performed as non-cycloplegic subjective refractions.

Anonymized patient spectacle lens sales data was provided by a major European manufacturer. This comprised lenses that had been manufactured and dispatched after an order was

received from a practitioner with the majority of lenses for delivery within Europe. The data was collated into histogram data using the SQLite database engine (Hipp, Wyrick & Company, Inc., Charlotte, North Carolina, USA) and analysed using the R statistical programming language. The data provided included the spherical power, cylindrical power and axis of the spectacle prescription. The lens design, diameter, laterality (prescribed for right or left eye) and date of manufacture were also included. For lens designs with an addition, this was also specified. The presence of an addition allowed the lenses to be separated into two groups, the single vision (SV) lens group and the addition (ADD) lens group. The data was validated for missing and malformed data fields and any lenses with incomplete or invalid data were excluded. The spherical equivalent power was calculated for each lens.

Data from the E3 study was extracted by digitizing the published results using Plot Digitiser [26]. Data from the GHS study [27], a population based observational study, was also digitized as an additional comparison. The GHS was chosen as an additional comparison as it took place in Germany, had a similar age range (35–74) and was one of the component studies of the E3 study. In addition, Germany was the largest contributor to the spectacle lens data.

Myopia was defined according to the International Myopia standards [28], with a spherical equivalent (SE) refractive error of $\leq$ -0.50 D being considered myopic, and $\leq$ -6.00 D considered highly myopic. Hyperopia was defined as $\geq$ +0.75 D and emmetropia defined as > -0.50 D and < +0.75 D. For comparison with the E3 study, analysis was also performed using the myopia definition used in that study, i.e. $\leq$ -0.75 D.

The E3 study, a meta-analysis on refractive error prevalence in Europe, was chosen as a comparative study for several reasons. Firstly, the manufacturer database reflected almost exclusively European lens sales. Secondly, as the spectacle lens data comprised a substantial proportion of reading addition lenses typically used by older presbyopic adults [29] (age $\geq$ 40–45 typically) [30], the adult age profile of the E3 consortium (age 25–89 years) was deemed suitable, and it was assumed that the datasets could be comparable. These age assumptions were also validated using the EMR data. With this more detailed optometric data, both the age and spectacle correction data were available, allowing determination of the age distribution of patients with single vision and reading addition spectacles. The relationship between age and reading addition was determined by fitting a logistic function to the age and right eye reading addition found in the EMR data using the 'drc' extension package for R [31]. A logistic function was also created to determine the number of individuals requiring a reading addition at each age from 1 to 100 years old within the EMR data. The base R predict function was then used to generate 95% prediction intervals for both logistic models. Probability density functions were generated for each reading addition value to determine the distribution of age associated with that reading addition. The ADD lens group then had an estimated age assigned for each spectacle lens based on the reading addition value for that lens using the probabilities generated from the EMR data.

The EMR data was randomly sampled to provide an age and gender matched population for comparison with the E3 population. The ADD lens data was also age matched with the E3 population using the estimated age for each lens. From the age matched EMR and ADD lens data, the proportion of myopia, high myopia and hyperopia present was calculated in 5-year age brackets to allow comparison with the E3 and GHS data.

## Results

### Spectacle lens dispensing and EMR refractive error distribution

The spectacle lens dataset comprised 141,547,436 lenses from the manufacturer sales records ranging from the year 1998 to 2016. The EMR dataset included 555,528 patient visits ranging

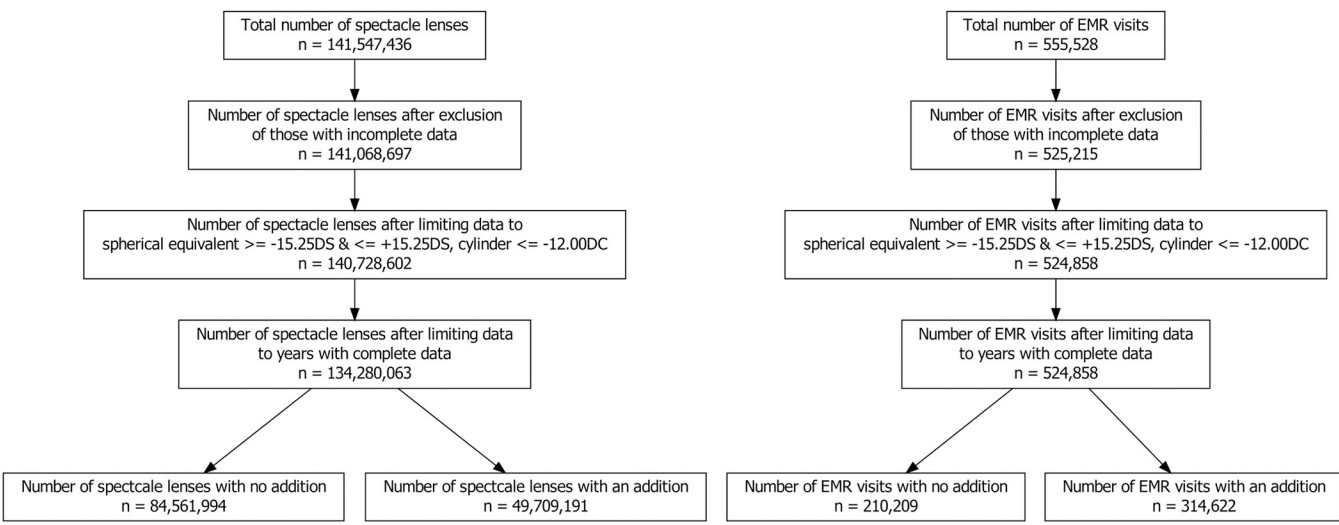

**Fig 1. Number of spectacle lenses and EMR visits included in analysis.**

from the year 1980 to 2019. Records with incomplete or missing data were excluded from both datasets and only years with complete data were included in the analysis (Fig 1). In total 134,280,063 spectacle lenses were included, comprised of 84,561,994 SV lenses and 49,709,191 ADD lenses. The final EMR dataset was composed of 524,868 patient visits.

Over 97% of spectacle lenses were for delivery within Europe with Germany accounting for the largest proportion (≈48%) of all lenses delivered. The EMR data included 244,002 unique patients representing 5.1% of the population of the Republic of Ireland [32]. The gender distribution of EMR patient visits was 51.3% female, 34.9% male and not recorded in 13.8% of records. The 28 optometric practices were located all across the Republic of Ireland representing both rural and urban populations.

The distribution of refractive error within the EMR data and spectacle lens data are presented in Fig 2, including the complete datasets and also segregated according to lens type (SV or ADD lens). Table 1 summarises the descriptive statistics for each distribution.

All distributions demonstrate the classic negatively skewed leptokurtotic curve found in most studies of refractive error, with the majority of observations centred close to emmetropia. The only exception to this pattern was the SV spectacle lenses which were found to have a bimodal distribution with a significant notch apparent at zero spherical equivalent.

### Estimating age using reading addition

Fig 3 shows the relationship between age and the presence of an addition by comparing the EMR distribution of SE for single vision prescriptions with those aged under 45 and the SE distribution of prescriptions with an addition and those aged 45 and over. It can be seen that the distribution of SE for those under age 45 (left panel, histogram bars) is very similar to the distribution of those prescribed a SV lens (left panel, dashed line), while the distribution of SE for those over age 45 (right panel, histogram bars) is very similar to the distribution of those prescribed an ADD lens (right panel, dashed line). The remarkable degree of similarity between being under age 45 and being prescribed single vision ($\chi^2$ = 552, p = 0.2365, DoF = 529) and being 45 years or older and being prescribed an addition ($\chi^2$ = 899, p = 0.2408, DoF = 870) indicates that age and the prescribing of an addition are highly correlated. Table 2 shows the relationship between age and the likelihood of prescribing a reading addition in the form of a

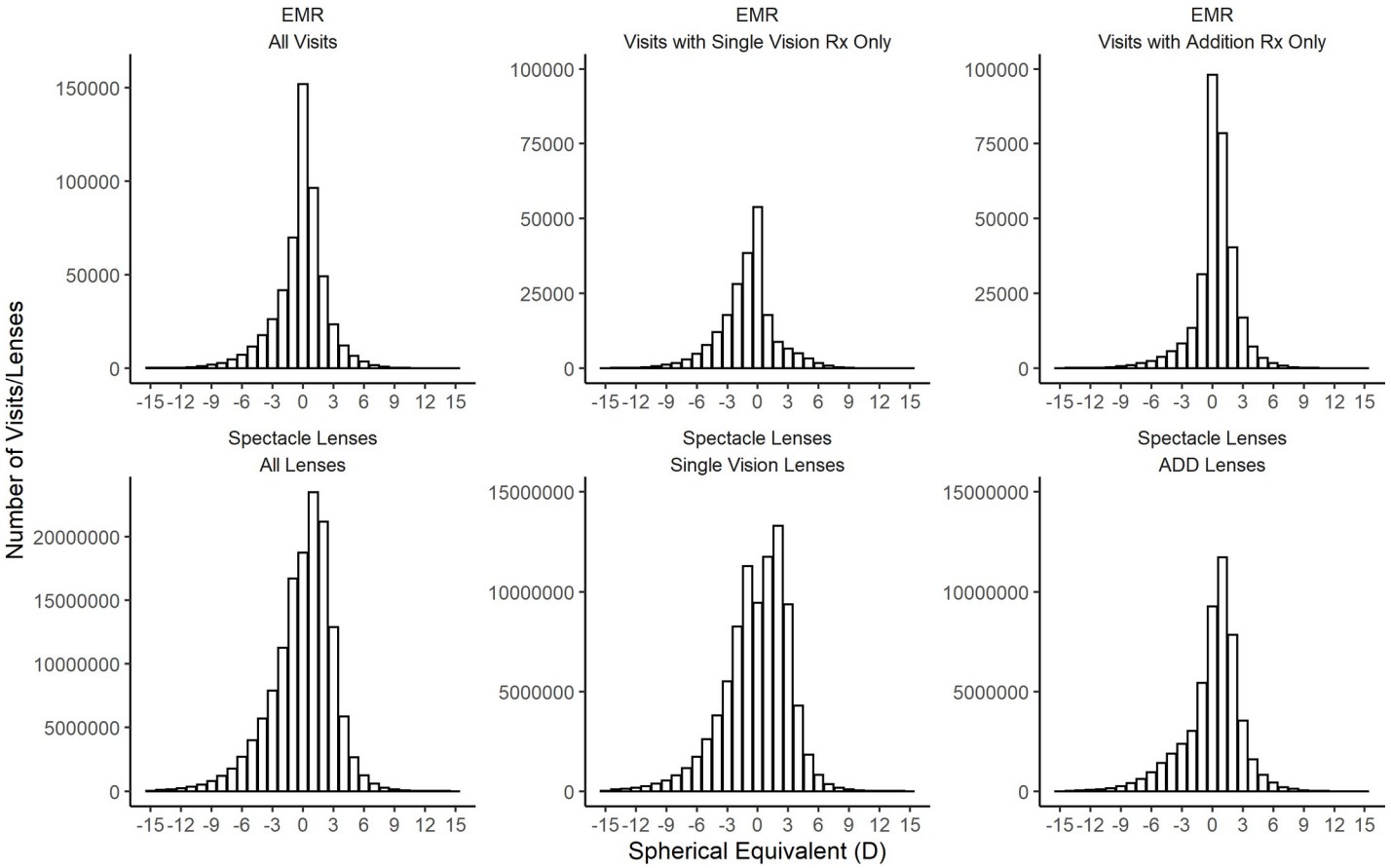

**Fig 2. Distribution of spherical equivalent in each dataset.** Top Panel—EMR data from Irish optometry practices. Right spherical equivalent distribution for all visits (n = 536,249), single vision prescriptions (n = 215,207) and addition prescriptions (n = 321,013). Bottom Panel—Spectacle Lens Distribution from manufacturer data for all lenses (n = 134,280,063), single vision, (SV) lenses (n = 84,561,994) and addition, (ADD) lenses (n = 49,709,191).

contingency table. A summary of the distributions and their statistical relationship is given in Table 3.

The relationship between age and the power of the addition given in glasses for the EMR data is shown in Fig 4. This relationship could be accurately fitted to a logistic function with nonlinear regression (estimate = 2.2 D, t = 818.94, p < 0.001). The residual standard error found was 7.56 years.

Fig 4 also shows the 95% prediction limits for estimating age if only the spectacle add power is known, as is the case with lens dispensing data. A logistic function was also fitted to the

**Table 1. Mean, range and distribution characteristics of spectacle lens and EMR data.**

| Dataset | Mean SE (D) ± SD | Skew | Kurtosis |
|---|---|---|---|
| All Spectacle Lenses | +0.02 ± 3.08 | -0.80 | 1.73 |
| SV Lenses | -0.03 ± 3.22 | -0.74 | 1.47 |
| ADD Lenses | +0.11 ± 2.84 | -0.89 | 2.20 |
| All EMR Visits | -0.13 ± 2.50 | -0.74 | 3.19 |
| Visits with SV Rx | -0.91 ± 2.74 | -0.30 | 2.09 |
| Visits with Add Rx | +0.39 ± 2.17 | -1.09 | 5.82 |

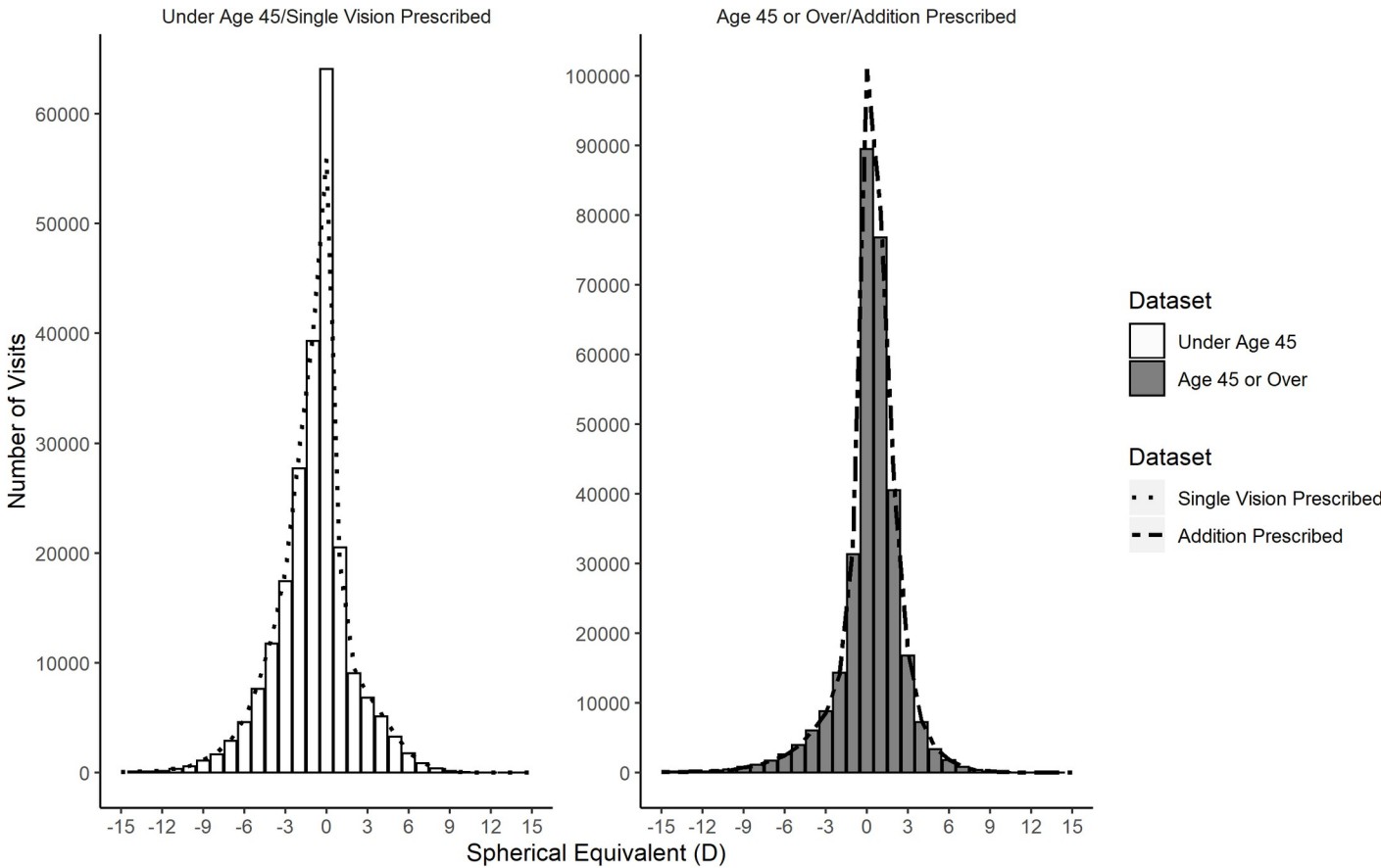

**Fig 3. Age and the prescribing of an addition are highly correlated in EMR patients.** Distribution of spherical equivalent for those under age 45 (left panel bars) and those age 45 and over (right panel bars). The dotted line represents the distribution of spherical equivalent for those given a single vision prescription (left panel) and those given a prescription containing an addition (right panel).

relationship between the probability of being prescribed a reading addition and age (estimate = 42.29 years, t = 653.73, p < 0.001). The residual standard error was 1.73%. This allows estimation of the proportion of individuals at each age likely to require a reading addition (Fig 5). These relationships were then used to infer ages for the ADD lens data. This allowed the generation of sub-populations of a given age for comparison with the EMR, E3 and GHS data. Using these two functions to determine age ranges and by generating probability density functions for each value of reading addition in the EMR data, the level of myopia, hyperopia and astigmatism was calculated for age groups from ≥45 years to ≤ 80 years for the ADD lens data.

## Comparison with E3

The distributions of spherical equivalent refraction in the E3 study and the age matched EMR data were closely matched ($\chi^2$ = 527, p = 0.29, DoF = 510) with both being negatively skewed leptokurtotic distributions (Fig 6).

**Table 2. Contingency table comparing the frequency of addition prescribing for EMR patients under age 45 and those age 45 and over.**

|  | No Addition Prescribed | Addition Prescribed |
|---|---|---|
| Under 45 | 204,027 | 24,512 |
| Age 45 or Over | 13,515 | 298,807 |

**Table 3. Descriptive statistics comparing single vision EMR prescriptions to younger EMR patients and addition EMR prescriptions to older EMR patients.**

| Dataset | Mean SE (D) | Skew | Kurtosis | Chi-Square Test |
|---|---|---|---|---|
| Single Vision | -0.91 ± 2.74 | -0.30 | 2.09 | $\chi^2 = 552$, p = 0.2365, DoF = 529 |
| Under Age 45 | -0.80 ± 2.66 | -0.30 | 2.26 | |
| Addition | +0.39 ± 2.17 | -1.09 | 5.82 | $\chi^2 = 899$, p = 0.2408, DoF = 870 |
| Over Age 45 | +0.36 ± 2.25 | -1.16 | 5.58 | |

Age-matched comparison of the level of myopia, hyperopia and astigmatism for EMR relative to E3 data revealed broadly similar distributions across the refractive error types, albeit that the distribution of myopia was lower and hyperopia higher in the EMR data relative to the E3 data (Table 4). The ADD lens data distributions of myopia, hyperopia and astigmatism were all higher but also similar to the age matched E3 data (Table 5).

The E3 reported levels of myopia, hyperopia and high myopia across various age groups were compared to the EMR, ADD lenses and GHS data across the same age groups (Figs 7–9). These figures show the EMR data is the closest match to the E3 data. Confidence intervals for the EMR data were found to be overlapping with the confidence intervals for E3 data at 7 age points for myopic refractions (Fig 7), 6 age points for hyperopic refractions (Fig 8) and 12 age points for highly myopic refractions (Fig 9). The ADD lens data, however, provides a closer

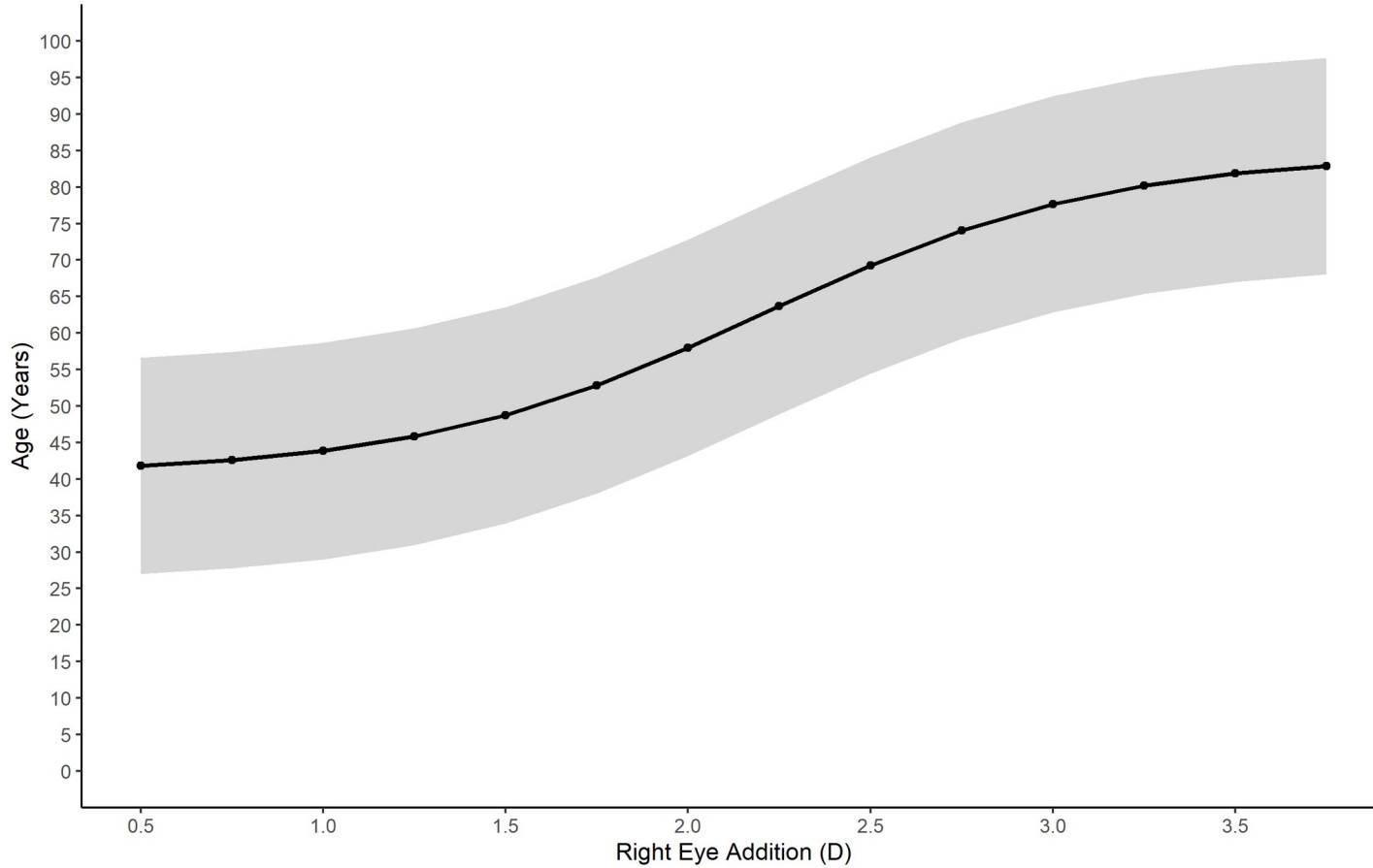

**Fig 4. Predicted age based on the prescribed reading addition for EMR patients with 95% prediction intervals.**

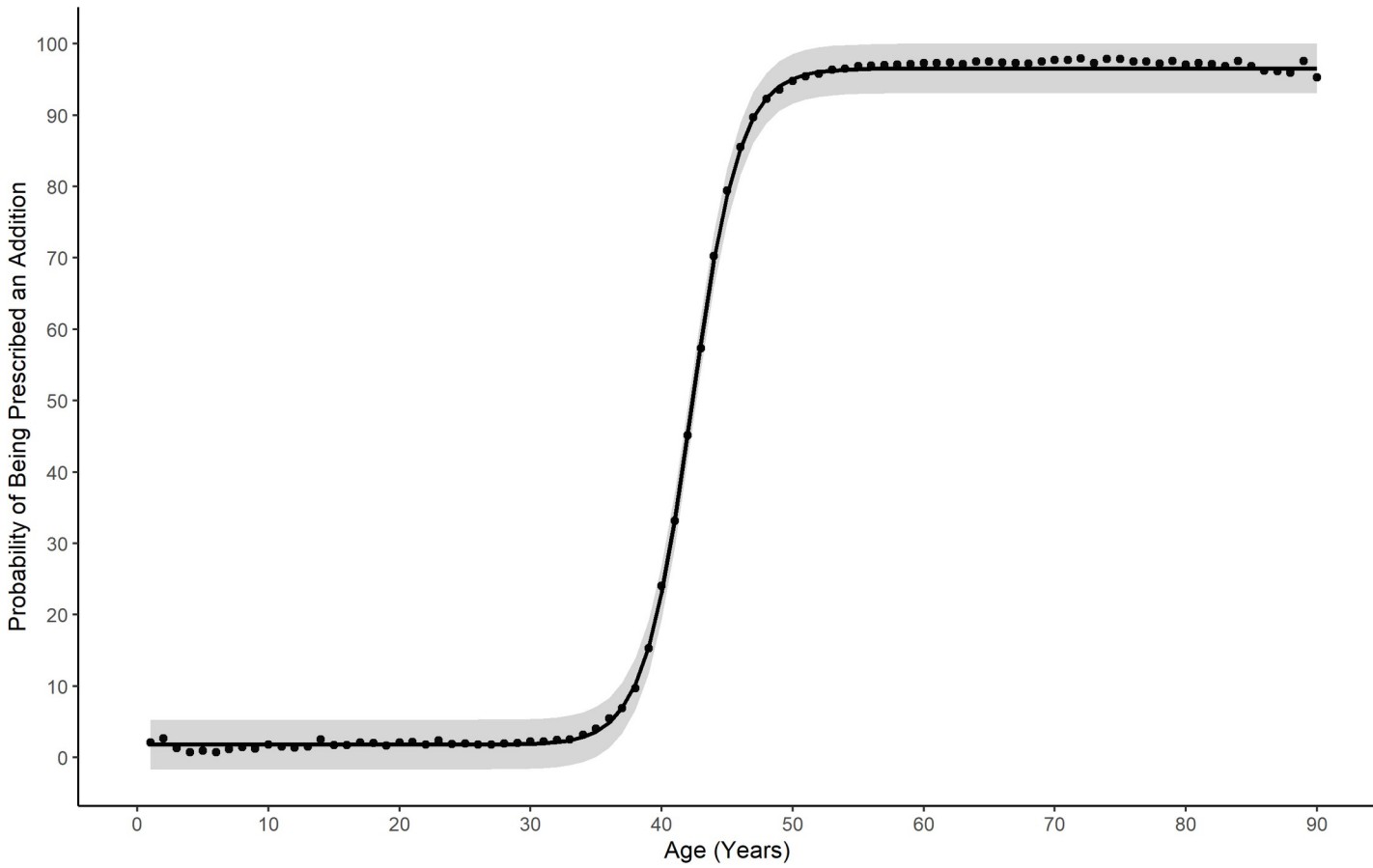

**Fig 5. Likelihood of needing a reading addition for EMR patients at different ages with 95% prediction intervals.**

approximation to the E3 data for total myopia compared to the GHS data, particularly up to age 64 (Fig 7).

## Discussion

Our results indicate that EMR data provides a close approximation to refractive error prevalence values found as part of the E3 study. Age related variation in the proportions of myopes and hyperopes are similar across the EMR and E3 data. Both the EMR and E3 datasets demonstrated high levels of myopia in younger age groups (Fig 7) which supports the findings of other studies demonstrating an increase in myopia prevalence in more recent generations [5,6]. Although the EMR data falls outside the E3 confidence intervals at some points for both the myopia and hyperopia comparisons, this is also true of the GHS data which was a component study of the E3 dataset, with the EMR data providing a closer match to the E3 than the GHS data. As the confidence intervals indicate the likely position of the mean of the study population some fluctuation is expected when comparing different study populations.

It was possible to estimate the likely recipient age for every spectacle lens prescription containing a reading addition by using the EMR data. This was achieved based on the observation that a significant majority of EMR patient visits below the age of 40 years were not prescribed an addition while the majority of patients visits above the age of 50 years were prescribed an addition. Along with the presence of an addition, the power of the reading addition was also

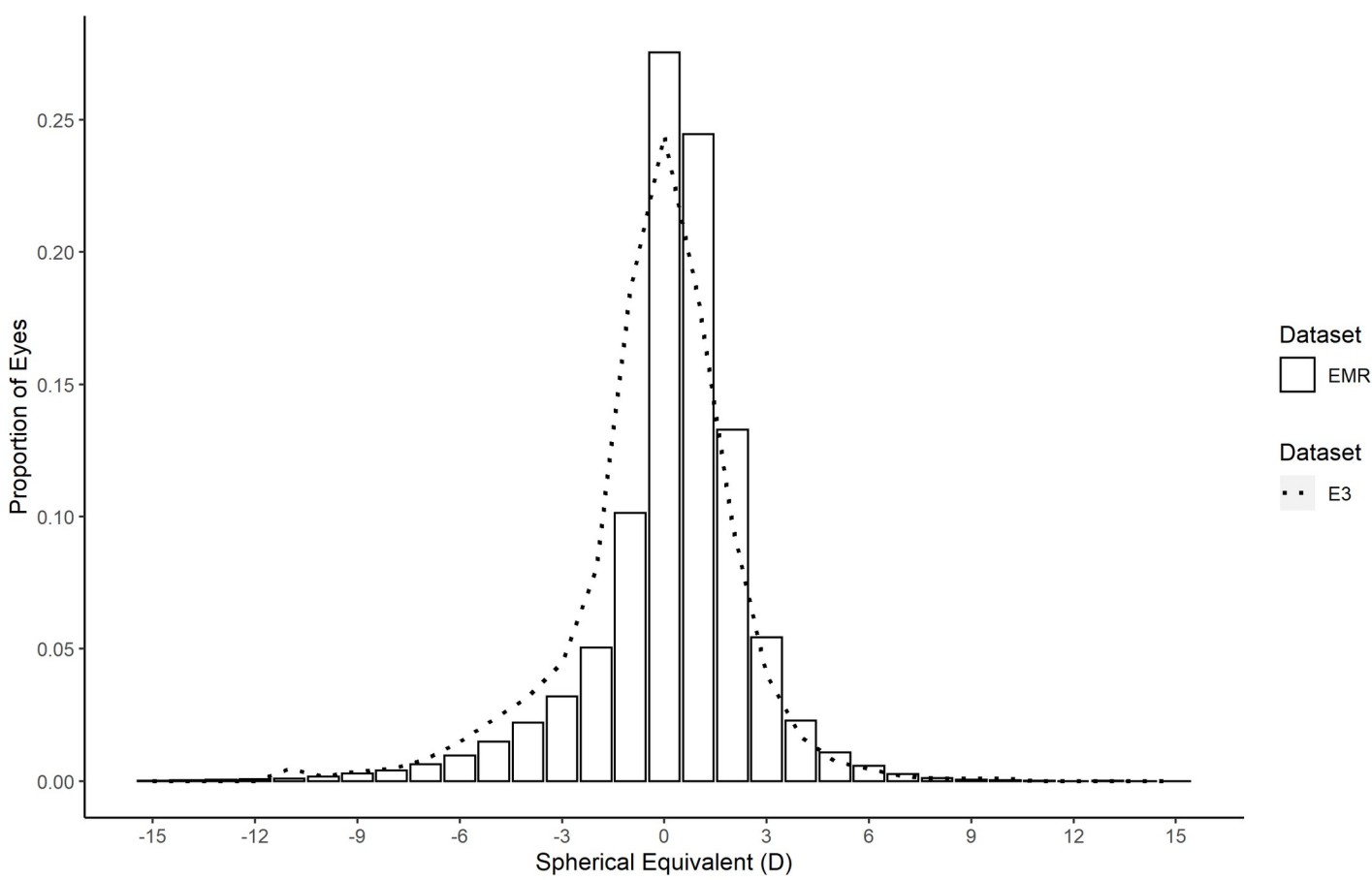

**Fig 6. Comparison of spherical equivalent distribution between E3 and EMR.** E3 distribution of refractive error spherical equivalent (dotted line) compared to the gender and age matched EMR distribution of right eye refractive error spherical equivalent (bars).

found to provide a means of estimating a patient's age. These inferences allowed an estimated age to be associated with each spectacle lens containing an addition within the spectacle lens sales dataset. The combination of disparate data sources to provide greater insight is a hallmark of Big Data analysis [33], and in this case allowed a deeper understanding of the usefulness of the spectacle lens sales data as a source of epidemiological data of refractive error.

Having accurate and current information on the prevalence of refractive error is vital to allow health services to plan for the increasing need for optical correction and the increased burden due to the ocular comorbidities [3,34–37] associated with increasing refractive error. Myopia is of particular concern as it is estimated that up to 49.8% of the global population will be myopic by 2050 and 9.8% of those will be highly myopic [4]. The combination of high myopia and increasing age have been found to be a risk factor for vision impairment and blindness

**Table 4. Age matched comparison of refractive error rates between the E3 consortium and EMR data (mean age = 60.16 ± 12.23 years).**

| Data Set | All Myopia ≤ -0.75 | Low Myopia ≤ -0.75 to > -3.00 | Moderate Myopia ≤ -3.00 to > -6.00 | High Myopia ≤ -6.00 | All Hyperopia ≥ +1.00 | High Hyperopia ≥ +3.00 | Emmetropia > -0.75 to < +1.00 | Astigmatism ≥ 1.00 |
|---|---|---|---|---|---|---|---|---|
| E3 (n = 62,393) | 30.60% | 19.50% | 8.08% | 2.71% | 25.23% | 5.37% | 44.17% | 23.86% |
| EMR (n = 200,076 | 21.52% | 13.56% | 5.70% | 2.26% | 37.89% | 7.38% | 40.59% | 28.38% |

**Table 5. Age matched comparison of refractive error rates between the E3 consortium and ADD lens data (mean age = 62.55 ± 8.59 years).**

| Data Set | All Myopia ≤ -0.75 | Low Myopia ≤ -0.75 to > -3.00 | Moderate Myopia ≤ -3.00 to > -6.00 | High Myopia ≤ -6.00 | All Hyperopia ≥ +1.00 | High Hyperopia ≥ +3.00 | Emmetropia > -0.75 to < +1.00 | Astigmatism ≥ 1.00 |
|---|---|---|---|---|---|---|---|---|
| E3 (n = 50,010) | 22.44% | 14.08% | 6.24% | 1.93% | 37.23% | 7.98% | 40.33% | 26.96% |
| ADD Lenses (n = 35,720,655 | 28.60% | 15.12% | 9.52% | 3.95% | 43.02% | 9.98% | 28.38% | 31.45% |

[38]. A recent meta-analysis found a significantly increased risk of myopic macular degeneration and retinal detachment in high myopes with reduced visual acuity and worse treatment outcomes in eyes with these conditions [39]. Assessing any change to the prevalence of high myopia within a population is the area of most concern when considering the ocular comorbidities associated with refractive error. EMR data contains refractive error information and patient demographics including age, which can help to determine the population risk of vision impairment. The EMR data provides a good match to the E3 study for high myopia (Fig 9) and as such may be an invaluable method to determine the ongoing risk of vision impairment.

While conventional epidemiological studies remain the gold standard, they have some disadvantages. The most reliable studies have large sample sizes allowing their results to be

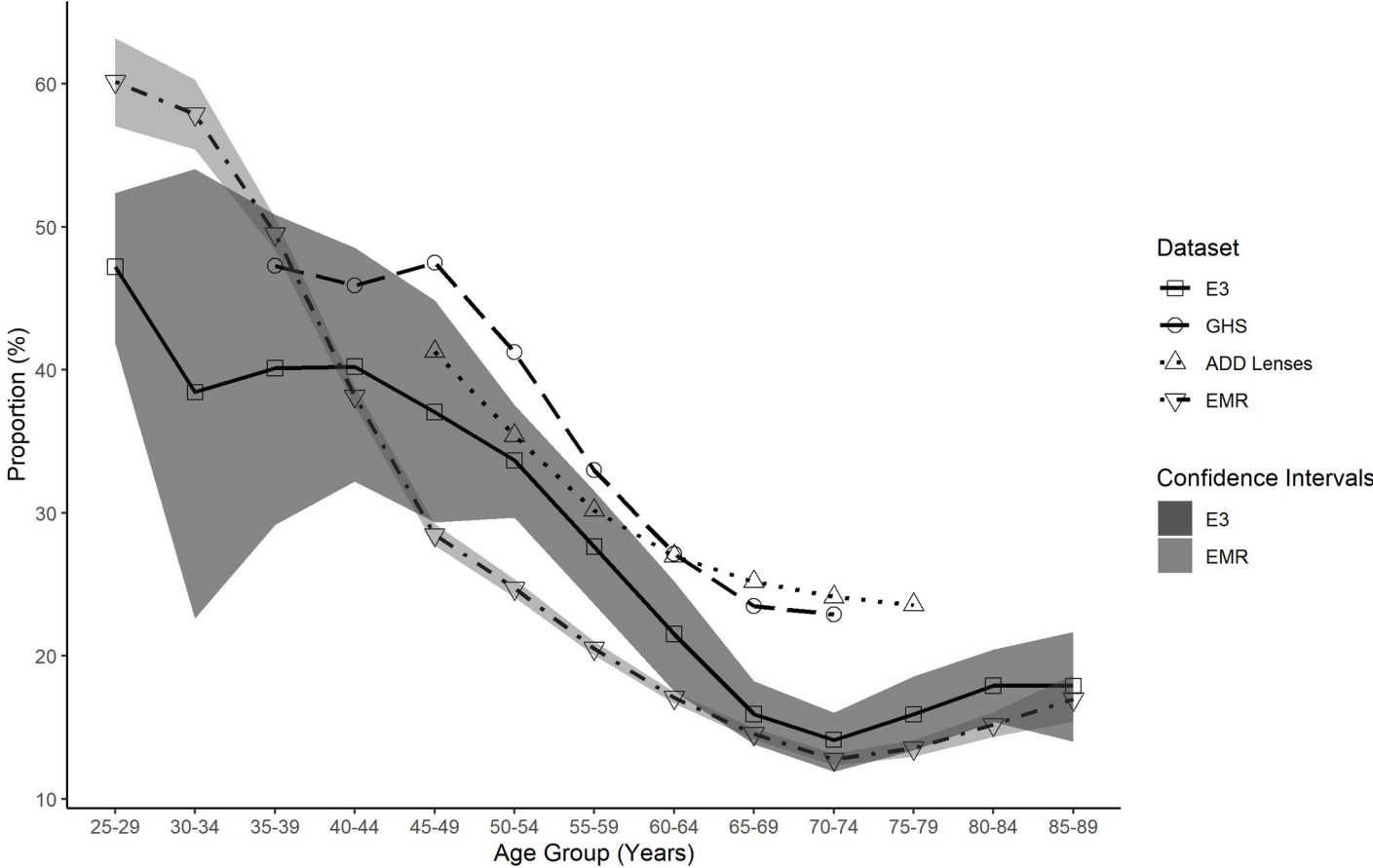

**Fig 7. Total myopia proportion for all data sets as a function of age group.** Total myopia proportion for EMR (inverted triangle), ADD Lenses (triangle), GHS (circle) and E3 (square) data as a function of age group. The E3 data confidence intervals (dark shaded area) are plotted to illustrate comparison with the other data sets. The EMR data confidence intervals (light shaded area) are plotted to show the overlap with the E3 data.

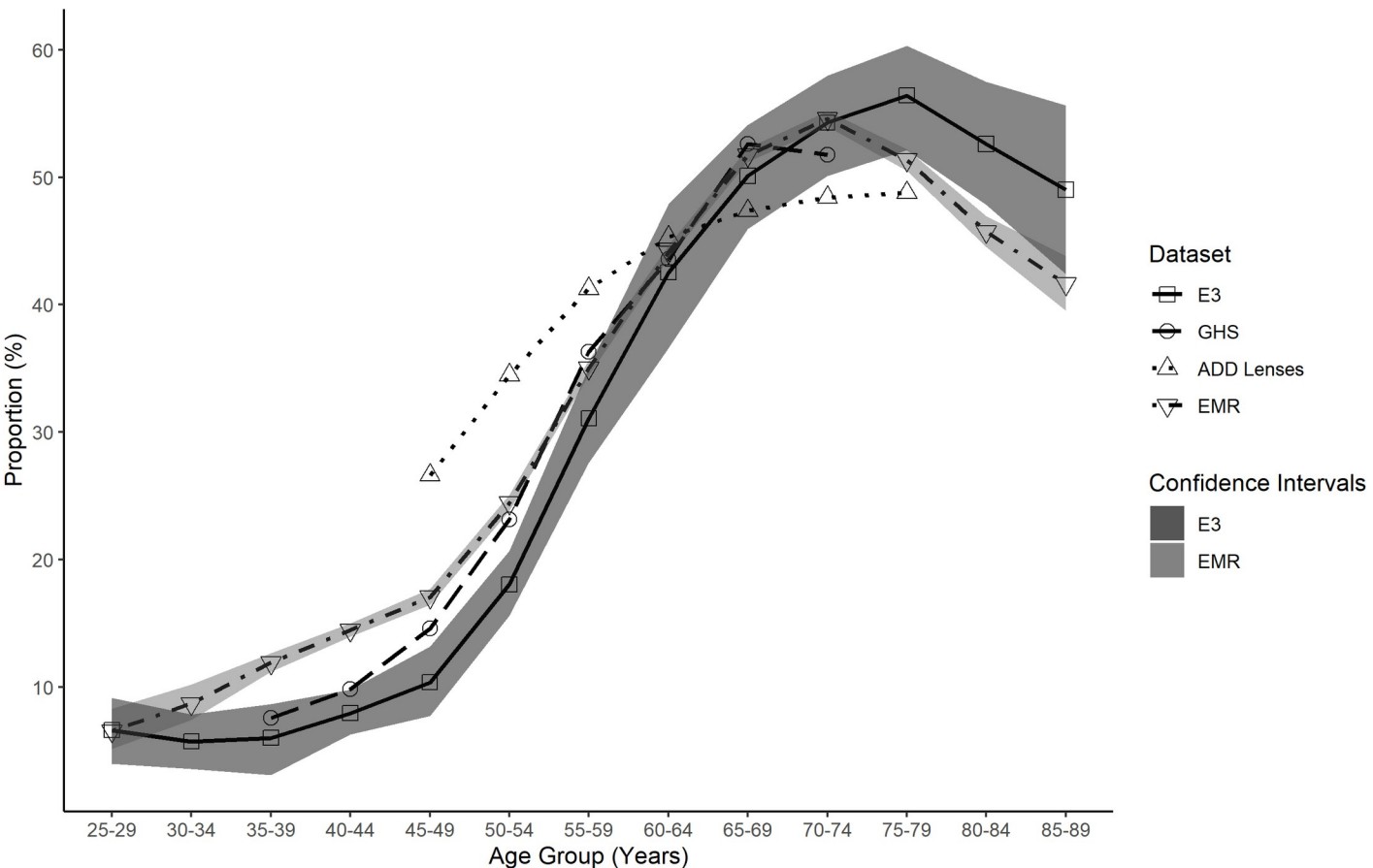

**Fig 8. Total hyperopia proportion for all data sets as a function of age group.** Total hyperopia proportion for EMR (inverted triangle), ADD Lenses (triangle), GHS (circle) and E3 (square) data as a function of age group. The E3 data confidence intervals (dark shaded area) are plotted to illustrate comparison with the other data sets. The EMR data confidence intervals (light shaded area) are plotted to show the overlap with the E3 data.

generalized to the entire population. Such sample sizes require significant investment and time to conduct the study, which perhaps explains the relative lack of epidemiological studies of refractive error and significant lack of longitudinal studies of refractive error. This paucity of data also contributes to uncertainty with regards to future projections of myopia prevalence [4]. Where such data is not available, EMR or industrial data may have a useful role as these are increasingly being collected as a matter of routine and can be collected with greater ease and at more regular intervals.

It is important to acknowledge that all epidemiological studies suffer from various forms of bias. For example, it is well established that most cross sectional studies suffer from volunteer bias, with volunteers usually from higher socio-economic backgrounds with a higher level of education [40]. Longitudinal studies frequently suffer from loss to follow up which may induce a bias in the profile of the remaining study population. It is important, therefore, when designing an epidemiological survey of refractive error to attempt to minimise these biases. Big data studies on refractive error will not suffer with the same biases as the data was not collected for the purpose of determining the population burden of refractive error. This type of epidemiological study will however, have a different set of biases which need to be considered. A frequent criticism of the secondary use of EMR data concerns the lack of access to healthcare of some population cohorts [41] due to a lack of health insurance. As this EMR data has come

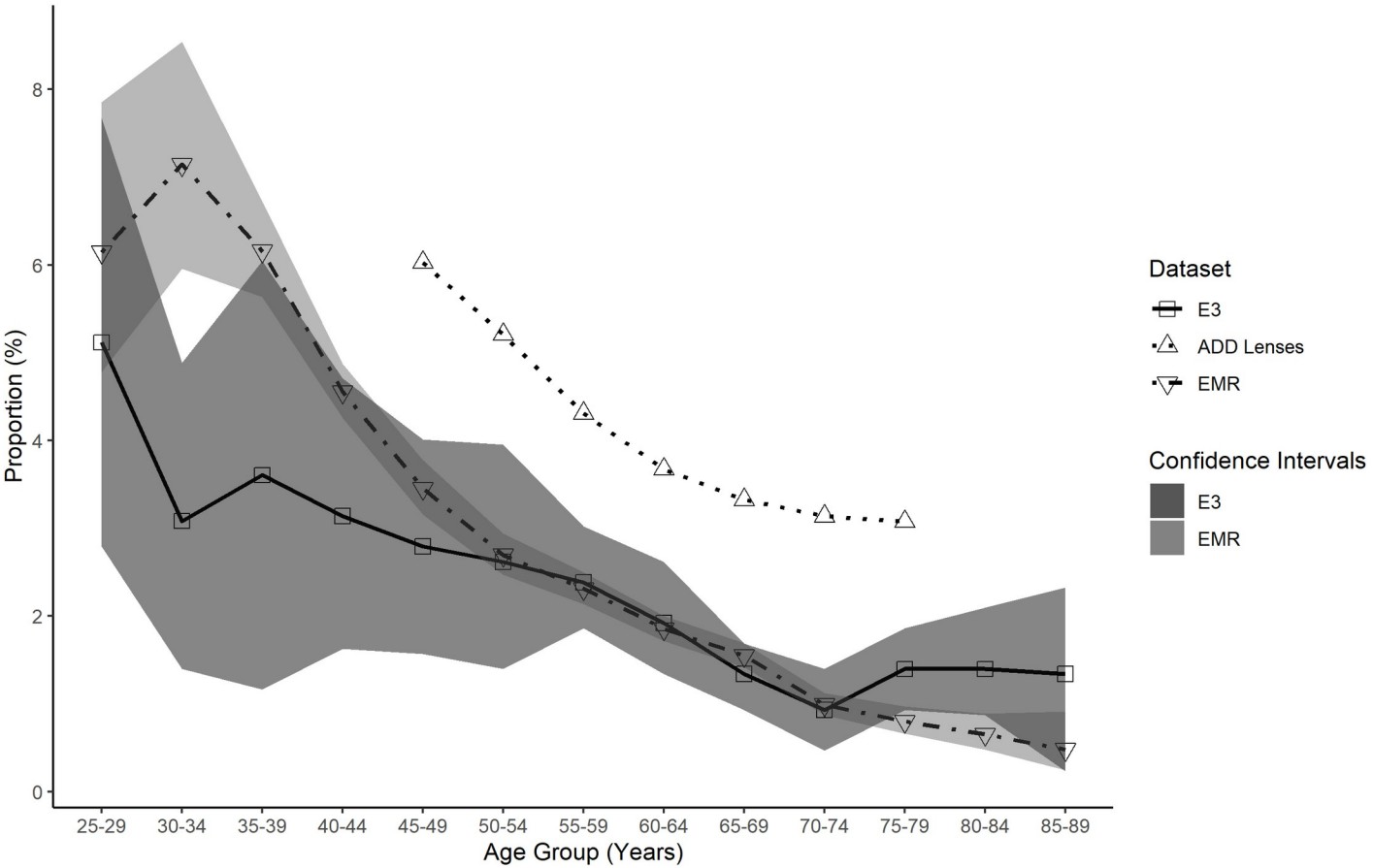

**Fig 9. Total high myopia proportion for all data sets as a function of age group.** Total high myopia proportion for EMR (inverted triangle), ADD Lenses (triangle) and E3 (square) data as a function of age group. The E3 data confidence intervals (dark shaded area) are plotted to illustrate comparison with the other data sets. The EMR data confidence intervals (light shaded area) are plotted to show the overlap with the E3 data. GHS not present as high myopia data was unavailable.

from a jurisdiction with free access to eyecare which is widely availed of, this should not create a significant bias in our data [42,43]. Less frequent replacement of spectacle lenses from those of lower socio-economic backgrounds may present a more significant issue with regards to the spectacle lens dispensing data. Measurement error can exist as a bias in any epidemiological study but may be well controlled in small studies through standardization of equipment and procedures. In a Big Data study of this nature, this is not possible. Nevertheless, error rates of subjective refraction in adults are typically low at between 1% and 2%, indicating the vast majority of refractions should be accurate to within ± 0.50 D of the correct refraction [44,45].

There are several limitations to this study that must be considered. In relation to spectacle lens data, demographic information of the individuals purchasing the spectacle lenses is not typically available in industrial datasets. Geographic information is likely to be available, however, which can provide some useful information. Using the EMR data to infer the age of a cohort of the spectacle lens users enhances the usefulness of this data, but the overall lack of demographic information means that further conclusions on subpopulations cannot be drawn. In this study, the spectacle lens data was supplied by one manufacturer. Economic factors and market penetration may have an effect on the background of the consumer choosing lenses from this manufacturer. Industrial data could be biased, for example, to particular socio-economic, ethnic or other demographic subgroups for reasons such as product cost,

geographic location and other factors specific to individual manufacturers. Higher educational attainment is associated with both socio-economic status and myopia [6], for example, so the possibility that the oversampling of individuals from particular backgrounds within individual datasets might influence population estimates of refractive error needs to be considered.

Under sampling of emmetropic patients is a more significant issue for the spectacle lens data as these represent spectacle lens sales. This will tend to produce an apparent increased proportion of hyperopic and myopic refractive errors, especially for younger subjects, as observed in this study. It is unlikely that emmetropic patients are purchasing spectacle lenses in significant numbers. This is particularly evident when considering the SV lenses in Fig 3. The notch apparent at zero dioptric power represents the reduction in purchasing of spectacle lenses by this group. It might be expected that the number of zero power lenses would be smaller than was observed, but there are plausible reasons to explain this. In cases of anisometropia one eye may have a zero-power lens when the fellow eye needs correction. In addition, the computation of spherical equivalent may result in zero spherical equivalent power for lenses prescribed to patients with mixed astigmatism. The lack of emmetropes represented within the spectacle lens sales data presents a problem and may explain the poorer match to the E3 study relative to EMR data. This implies that such data may be more representative of the distribution of refractive error within a population above a certain threshold of refractive error. The greatest risks of visual impairment are associated with high levels of myopia [39], and also high levels of hyperopia [3], both categories likely to seek optical correction. Further analysis and modelling may remove the limitation associated with the under sampling of emmetropes and allow the determination of the risk of vision impairment in those using spectacle lenses to correct higher refractive errors.

There are less limitations applicable to the EMR data due to the increased demographic detail captured in this data. Under sampling of emmetropic patients is likely to be less problematic for the EMR data which includes refraction data found as part of a patient's eye examination. Emmetropic patients are still likely to attend routine eye examinations for the purposes of screening for common ocular pathologies such as glaucoma and cataract [46] although some under sampling of young emmetropic patients may have still occurred. Importantly, EMR data is likely to be highly representative of the older population given the almost universal need for optical correction as presbyopia begins to manifest as a problem, even for emmetropes and low hyperopes who did not previously need correction. This is particularly the case in most countries in Europe where subsidised eye examinations are accessible to the majority of the population [47]. The close match of the EMR and E3 data observed herein suggests that the EMR is representative of the population at large.

In this EMR dataset, it was not possible to tell what type of refraction had been performed to reach the refractive error prescribed. Cycloplegic refraction is performed to avoid the errors in refraction that can be induced by accommodation in children and the use of cycloplegia is considered the most appropriate method to assess refractive error for research purposes [48]. Although it is unknown how many of these refractions have been performed with the aid of cycloplegia, a significant number of epidemiological surveys on refractive error have been carried out without the use of cycloplegia [7]. It has been found that accommodation mostly affects the determination of refractive error in children and has little impact on adults [49,50], particularly older adults [51]. The technique of refraction used, therefore, should have little impact on the primarily adult dataset used herein.

## Conclusion

The prevalence of refractive error within a population can be estimated using EMR data in the absence of population surveys. Results from EMR data also allow age to be inferred from the

addition in a spectacle lens. Industry derived sales can then be used to provide insights on the epidemiology of refractive errors in a population over certain age ranges. EMR and industrial data may therefore provide a fast and cost-effective surrogate measure of refractive error distribution that can be used for future health service planning purposes.

## Author Contributions

**Conceptualization:** Michael Moore, James Loughman, Daniel I. Flitcroft.

**Data curation:** Michael Moore.

**Formal analysis:** Michael Moore, John S. Butler, Daniel I. Flitcroft.

**Investigation:** Michael Moore, Daniel I. Flitcroft.

**Methodology:** Michael Moore, Daniel I. Flitcroft.

**Project administration:** James Loughman.

**Resources:** James Loughman, Arne Ohlendorf, Siegfried Wahl.

**Software:** Michael Moore, Daniel I. Flitcroft.

**Supervision:** James Loughman, John S. Butler, Daniel I. Flitcroft.

**Validation:** Michael Moore, John S. Butler, Daniel I. Flitcroft.

**Visualization:** Michael Moore.

**Writing – original draft:** Michael Moore.

**Writing – review & editing:** James Loughman, John S. Butler, Arne Ohlendorf, Siegfried Wahl, Daniel I. Flitcroft.

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
