## [Decision Letter · Decision Letter 0]

22 Feb 2021

PONE-D-20-40228

Application of Big-Data for Epidemiological Studies of Refractive Error

PLOS ONE

Dear Dr. Moore,

Thank you for submitting your manuscript to PLOS ONE. After careful consideration, we feel that it has merit but does not fully meet PLOS ONE’s publication criteria as it currently stands. Therefore, we invite you to submit a revised version of the manuscript that addresses the points raised during the review process.

We look forward to receiving your revised manuscript.

Kind regards,

Michael Mimouni

Academic Editor

PLOS ONE

Journal Requirements:

"I have read the journal's policy and the authors of this manuscript have the following competing interests: Arne Ohlendorf and Siegfried Wahl are employees of Carl Zeiss Vision International GmbH."

We note that one or more of the authors are employed by a commercial company: Carl Zeiss Vision International GmbH.

2.1. Please provide an amended Funding Statement declaring this commercial affiliation, as well as a statement regarding the Role of Funders in your study. If the funding organization did not play a role in the study design, data collection and analysis, decision to publish, or preparation of the manuscript and only provided financial support in the form of authors' salaries and/or research materials, please review your statements relating to the author contributions, and ensure you have specifically and accurately indicated the role(s) that these authors had in your study. You can update author roles in the Author Contributions section of the online submission form.

2.2. Please also provide an updated Competing Interests Statement declaring this commercial affiliation along with any other relevant declarations relating to employment, consultancy, patents, products in development, or marketed products, etc.  

4. Please amend your manuscript to include your abstract after the title page.

Reviewers' comments:

Reviewer's Responses to Questions

**Comments to the Author**

1. Is the manuscript technically sound, and do the data support the conclusions?

Reviewer #1: Yes

Reviewer #2: Yes

2. Has the statistical analysis been performed appropriately and rigorously? 

Reviewer #1: Yes

Reviewer #2: Yes

3. Have the authors made all data underlying the findings in their manuscript fully available?

Reviewer #1: Yes

Reviewer #2: No

4. Is the manuscript presented in an intelligible fashion and written in standard English?

Reviewer #1: Yes

Reviewer #2: Yes

5. Review Comments to the Author

Reviewer #1: This is an interesting paper which explored the validity of using electronic medical records (EMR) and spectacle lens sales data to estimate the prevalence and distribution of refractive errors in the population. The use of medical records to conduct various epidemiological studies is a growing trend and the big data approach can indeed provide a very valuable information, which can be used for planning purposes by national and local governmental organizations. The paper uses appropriate statistical analyses and is generally very well written. Conclusions are also generally well supported by the data. I only have a few minor comments, which should be addressed before the paper is published.

1. Limitations of the study are generally well considered; however, I feel that some of the statements related to the use of the EMR data should be made more balanced. The authors should carefully consider the limitations of using the EMR data to obtain a measure of the prevalence and distribution of refractive errors in the population. I somewhat disagree with the statement that “Undersampling of emmetropic patients is likely to be less problematic for the EMR data which includes refraction data found as part of a patient’s eye examination.” The undersampling of emmetropes is likely to be quite prominent in young adults who are unlikely to visit an optometry practice unless they have myopia. Interestingly, EMR dataset shows higher prevalence of myopia among young individuals compared to E3. I was left with the impression that I could not decide whether this was a result of a genuinely higher prevalence of myopia in Ireland compared to the overall European sample, or this was a result of a bias cause by the undersampling of emmetropes.

2. Both EMR and E3 datasets show sharp increase in the prevalence of myopia in younger generations 25-29 compared to other age groups, which is important to emphasize.

3. Please, clearly identify in the text and figure/table legends what dataset is being analyzed/discussed.

Reviewer #2: This is a study regarding application of big-data for epidemiological studies of refractive error . The discussion section is very well written, and the figures are very clear, therefore I do not have any questions to the authors.

6. PLOS authors have the option to publish the peer review history of their article (what does this mean?). If published, this will include your full peer review and any attached files.

Reviewer #1: No

Reviewer #2: No

---

## [Author Response · Author response to Decision Letter 0]

12 Mar 2021

Dear Dr. Mimouni, 

Thank you for requesting us to submit a revised draft of our manuscript entitled “Application of Big-Data for Epidemiological Studies of Refractive Error” to PLOS ONE. We sincerely appreciate the time and effort undertaken by you and each of the reviewers when considering our manuscript and in providing your feedback. We are delighted to resubmit our manuscript for further consideration having incorporated changes that reflect the comments you have provided. 

To facilitate your review of our revisions, the following is a point-by-point response to the questions and comments delivered in your email of 22/02/2021.

Editor Suggestions:

1. Please ensure that your manuscript meets PLOS ONE's style requirements, including those for file naming

Thank you for your feedback. We have made edits to the title page and manuscript to ensure they conform to PLOS ONE’s style requirements

2. Please provide an amended Funding Statement declaring this commercial affiliation, as well as a statement regarding the Role of Funders in your study

This has been updated on the submission portal and in the accompanying cover letter

3. Please also provide an updated Competing Interests Statement declaring this commercial affiliation along with any other relevant declarations relating to employment, consultancy, patents, products in development, or marketed products, etc.

This has been updated on the submission portal and in the accompanying cover letter

This has been updated in the accompanying cover letter

Reviewer 1 comments:

1. Limitations of the study are generally well considered; however, I feel that some of the statements related to the use of the EMR data should be made more balanced. The authors should carefully consider the limitations of using the EMR data to obtain a measure of the prevalence and distribution of refractive errors in the population. I somewhat disagree with the statement that “Undersampling of emmetropic patients is likely to be less problematic for the EMR data which includes refraction data found as part of a patient’s eye examination.” The undersampling of emmetropes is likely to be quite prominent in young adults who are unlikely to visit an optometry practice unless they have myopia. Interestingly, EMR dataset shows higher prevalence of myopia among young individuals compared to E3. I was left with the impression that I could not decide whether this was a result of a genuinely higher prevalence of myopia in Ireland compared to the overall European sample, or this was a result of a bias cause by the undersampling of emmetropes.

Thank you for your feedback. We have edited this point to take the reviewers comment into account:

“There are less limitations applicable to the EMR data due to the increased demographic detail captured in this data. Under sampling of young emmetropic patients may still present an issue for the EMR data which includes refraction data found as part of a patient’s eye examination. Young patients without significant refractive error are less likely to attend an optometry practice and this may explain the higher levels of myopia observed in young age groups when compared to the E3 data (Fig 7). Despite this, EMR data is still likely more representative than spectacle lens data for young patients as some will still attend for the purposes vision screening for occupational requirements and driving licensure and for screening of common ocular pathologies such as glaucoma and cataract [46].”

2. Both EMR and E3 datasets show sharp increase in the prevalence of myopia in younger generations 25-29 compared to other age groups, which is important to emphasize.

We agree this point should be emphasised and have added the following point in the first paragraph of the discussion:

“Both the EMR and E3 datasets demonstrated high levels of myopia in younger age groups (Fig 7) which supports the findings of other studies demonstrating an increase in myopia prevalence in more recent generations [5,6].”

As mentioned in point 1 above we have also more clearly indicated there is some uncertainty with this finding later in the discussion:

“Young patients without significant refractive error are less likely to attend an optometry practice and this may explain the higher levels of myopia observed in young age groups when compared to the E3 data (Fig 7).”

3. Please, clearly identify in the text and figure/table legends what dataset is being analyzed/discussed.

The text and figure/table legends have been edited to more clearly identify the dataset being discussed

Thank you for giving us the opportunity to improve our work through your valuable feedback. We have tried to incorporate your comments and hope these revisions will encourage you to accept our submission.

Yours Sincerely,

Michael Moore

---

## [Decision Letter · Decision Letter 1]

7 Apr 2021

Application of big-data for epidemiological studies of refractive error

PONE-D-20-40228R1

Dear Dr. Moore,

We’re pleased to inform you that your manuscript has been judged scientifically suitable for publication and will be formally accepted for publication once it meets all outstanding technical requirements.

Kind regards,

Michael Mimouni

Academic Editor

PLOS ONE

Additional Editor Comments (optional):

Reviewers' comments:

Reviewer's Responses to Questions

**Comments to the Author**

1. If the authors have adequately addressed your comments raised in a previous round of review and you feel that this manuscript is now acceptable for publication, you may indicate that here to bypass the “Comments to the Author” section, enter your conflict of interest statement in the “Confidential to Editor” section, and submit your "Accept" recommendation.

Reviewer #1: All comments have been addressed

Reviewer #2: All comments have been addressed

2. Is the manuscript technically sound, and do the data support the conclusions?

Reviewer #1: Yes

Reviewer #2: Yes

3. Has the statistical analysis been performed appropriately and rigorously? 

Reviewer #1: Yes

Reviewer #2: Yes

4. Have the authors made all data underlying the findings in their manuscript fully available?

Reviewer #1: Yes

Reviewer #2: No

5. Is the manuscript presented in an intelligible fashion and written in standard English?

Reviewer #1: Yes

Reviewer #2: Yes

6. Review Comments to the Author

Reviewer #1: All my comments were addressed. The manuscript is addressing an important question. I recommend acceptance.

Reviewer #2: All the questions have been addressed by the authors. I do not have any more comments or questions.

7. PLOS authors have the option to publish the peer review history of their article (what does this mean?). If published, this will include your full peer review and any attached files.

Reviewer #1: **Yes: **Andrei V. Tkatchenko

Reviewer #2: No

---

## [Editor Report · Acceptance letter]

12 Apr 2021

PONE-D-20-40228R1 

Application of big-data for epidemiological studies of refractive error 

Dear Dr. Moore:

I'm pleased to inform you that your manuscript has been deemed suitable for publication in PLOS ONE. Congratulations! Your manuscript is now with our production department. 

Kind regards, 

on behalf of

Dr. Michael Mimouni 

Academic Editor

PLOS ONE